# Identification of Novel Transcriptome Signature as a Potential Prognostic Biomarker for Anti-Angiogenic Therapy in Glioblastoma Multiforme

**DOI:** 10.3390/cancers13051013

**Published:** 2021-03-01

**Authors:** Shuhua Zheng, Wensi Tao

**Affiliations:** 1College of Osteopathic Medicine, Nova Southeastern University, Fort Lauderdale, FL 33314, USA; 2Department of Radiation Oncology, University of Miami-Miller School of Medicine, Miami, FL 33136, USA

**Keywords:** glioblastoma multiforme, *SOCS3*, *VEGFA*, angiogenesis, bevacizumab

## Abstract

Glioblastoma multiforme (GBM) is the most common and devastating type of primary brain tumor, with a median survival time of only 15 months. Having a clinically applicable genetic biomarker would lead to a paradigm shift in precise diagnosis, personalized therapeutic decisions, and prognostic prediction for GBM. Radiogenomic profiling connecting radiological imaging features with molecular alterations will offer a noninvasive method for genomic studies of GBM. To this end, we analyzed over 3800 glioma and GBM cases across four independent datasets. The Chinese Glioma Genome Atlas (CGGA) and The Cancer Genome Atlas (TCGA) databases were employed for RNA-Seq analysis, whereas the Ivy Glioblastoma Atlas Project (Ivy-GAP) and The Cancer Imaging Archive (TCIA) provided clinicopathological data. The Clinical Proteomic Tumor Analysis Consortium Glioblastoma Multiforme (CPTAC-GBM) was used for proteomic analysis. We identified a simple three-gene transcriptome signature—*SOCS3*, *VEGFA*, and *TEK*—that can connect GBM’s overall prognosis with genes’ expression and simultaneously correlate radiographical features of perfusion imaging with *SOCS3* expression levels. More importantly, the rampant development of neovascularization in GBM offers a promising target for therapeutic intervention. However, treatment with bevacizumab failed to improve overall survival. We identified *SOCS3* expression levels as a potential selection marker for patients who may benefit from early initiation of angiogenesis inhibitors.

## 1. Introduction

Despite intensive research efforts and continuous advances in treatment options, glioblastoma multiforme (GBM) remains the most common and aggressive type of tumor in the central nervous system (CNS) in adults [1]. The Central Brain Tumor Registry of the United States (CBTRUS) reported that the incidence rate of malignant CNS tumors in the United States is 23.41 per 100,000, 48.3% of which are GBM [2]. Molecular profiling (World Health Organization [WHO] grades II–III) has classified lower-grade gliomas (LLGs) into three distinct molecular subgroups: isocitrate dehydrogenase (*IDH*) wildtype, *IDH*-mutated with 1p/19q codeletion, and *IDH*-mutated without 1p/19q codeletion [3]. However, such molecular profiling for GBM is still lacking [4]. Therefore, strategies for improving GBM prognosis are urgently needed. Numerous studies using stringent screening algorithms to generate multiple random prognostic transcriptome signatures usually lacked functional interpretation, restricting the clinical applicability and therapeutic significance of these signatures [5,6,7,8,9,10]. Recent studies identified overactivation of the Cullin-really interesting new gene (RING) E3 ligase (CRL) in the ubiquitin-proteasomal system (UPS) as an unfavorable factor for patients with GBM [11]. Recent investigation revealed that elevated expression levels of the substrate-binding protein suppressor of cytokine signaling 3 (SOCS3) in the CRL5 complex are correlated with chemo- and radioresistance in GBM [12,13,14]. However, the underlying molecular functional mechanisms of SOCS3 in GBM progression remain largely unknown.

Previously, our group and others identified the perivascular localization of SOCS3 in GBM samples [13]. Microvascular proliferation is often observed in high-grade malignant gliomas and GBM with poor prognosis [15]. The vascular structure of GBM is characterized by abundant immature endothelial cells with loose conjunction, fenestrated structure, and a discontinuous membrane that create large-caliber and aberrant vascular walls [16]. However, expression levels of angiogenic markers, including hypoxia-inducible factor 1α (HIF-1α), HIF2α, vascular endothelial growth factor (VEGF), VEGF receptor 1 (VEGFR1), and VEGFR2, usually cannot reliably predict GBM prognosis or therapeutic outcomes, possibly due to the rapid turnover of angiogenic proteins [17]. Degradation of HIF-1α is rapidly executed by von Hippel–Lindau (VHL)-mediated ubiquitination and subsequent degradation, whereas the ubiquitin conjugation of VHL is mediated by SOCS3 in the CRL5 E3 ligase complex [18]. As such, it is worthwhile to investigate the potential involvement of SOCS3 in regulating angiogenic activities and as an independent prognostic factor of treatment outcomes in GBM.

## 2. Results

### 2.1. Anatomically Mapped Differential Gene Expression of SOCS3 and Angiogenesis Markers

*SOCS3* and the angiogenesis markers *ANGPT1*, *ANGPT2*, *FLT1*, *PECAM1*, *TEK*, *TIE1*, *VEGFA*, *NRP1*, and *KDR* were classified anatomically into structures of leading edge (LE), infiltrating tumor (IT), cellular tumor (CT), pseudopalisading cells around necrosis (PAN), and microvascular proliferation (MVP) from the histological Ivy Glioblastoma Atlas Project (Ivy-GAP) dataset (Figure 1A) [19]. Most angiogenesis markers, including *ANGPT1*, *ANGPT2*, *FLT1*, *PECAM1*, *TEK*, *TIE1*, *NRP1*, and *KDR*, were identified as significantly enriched in the MVP area (Figure 1A). The distribution of *VEGFA* expression was mainly found in the PAN area, whereas *SOCS3* expression was primarily identified in the PAN and MVP areas (Figure 1A). These patterns of distribution were further confirmed by in situ hybridization (ISH) data showing higher *SOCS3* expression in perivascular and PAN areas, whereas *VEGFA* expression was mainly found in the PAN area (Figure 1B). ISH analysis of normal murine cranial tissue demonstrated perivascular *TEK* expression. These data suggested that SOCS3’s involvement in angiogenic activities might be important in human glioblastoma carcinogenesis.

### 2.2. Elevated SOCS3 Protein Levels and GBM Neovascularization

SOCS3 functions as an active substrate recruitment modular protein in CRL5, which mediates the degradation of the anti-angiogenic tumor suppressor protein VHL [18] (Figure 1C). From the proteomic study in the CPTAC dataset, elevated protein levels of cullin5, the scaffold protein of the CRL5 complex, correlated negatively with VHL protein levels (Figure 2A). Given the potent anti-angiogenic activities of the VHL protein, elevated SOCS3 in the CRL5 complex may contribute to the higher angiogenesis in samples with high SOCS3 protein levels. Furthermore, we investigated the vascularization phenotypes in GBM cases with differential SOCS3 staining intensities using immunohistochemistry (IHC), ISH and hematoxylin and eosin (H&E) staining analysis. We found a positive correlation in samples with stronger SOCS3 immunoreactivity in perivascular area and higher blood vessel densities (Figure 2B, right panels) compared to those with weaker perivascular SOCS3 staining intensities (Figure 2B, left panels). We also found that samples with higher perivascular SOCS3 expression in ISH analysis generally have more enriched neovascularization (Appendix A). To further test the potential role of SOCS3 in angiogenesis, GBM samples with differential expression levels of *SOCS3* were selected from the TCGA-GBM dataset and analyzed for blood vessel invasion and H&E staining analysis. Again, samples with expression levels of *SOCS3* higher than the median are often more vascularized than samples with *SOCS3* expression lower than the median (Appendix A). All these data suggested that elevated *SOCS3* expression is positively correlated with increased neovascularization.

### 2.3. Prognostic Index (PI) System-Based Three-Gene Signature for GBM Prognosis

Microvascular proliferation is a key feature of glioma grading and progression of malignant GBM [15]. However, expression levels of angiogenic markers cannot accurately predict GBM prognosis [17]. Verhaak et al. classified GBM into four distinct cellular types: classical, mesenchymal, proneural, and neural (C, M, P and N, respectively) [20]. *TEK* and *VEGFA* were selected in the three-gene transcriptomic signature due to their endothelial cell-specific functionalities [21] (Figure 3A). Based on the GBM-BioDP dataset using RNA-Seq data of *SOCS3*, *VEGFA*, and *TEK* as covariates, we evaluated the prognostic index (PI) for the three-gene transcriptome signature in the subclasses of GBM. We found that the three-gene signature has PI values of 1.52, 1.86, 2.32, 4.01, and 6.65 for the GBM full cohort and the C, M, P, and N subclasses, respectively (Figure 3B). *P*-value analysis for overall survival (OS) showed 0.008, 0.04, 0.005, <0.0001, and 0.02 for the GBM full cohort and the C, M, P, and N subclasses, respectively, using the three-gene signature in grouping GBM patients into high-risk or low-risk groups based on the median cutoff value. These data indicated that the three-gene signature composed of *SOCS3*, *VEGFA*, and *TEK*, encoding SOCS3, angiogenic cytokine VEGF-A, and the TEK receptor, respectively, can be used independently to predict prognosis of all GBM subclasses.

### 2.4. Relationship between SOCS3, VEGFA, and TEK Expression and the Status of IDH Mutations

*IDHs* mutation can regulate angiogenesis via either HIF-1α stabilization or altered expression of angiogenic proteins [22,23]. Analysis of the Chinese Glioma Genome Atlas Network (CGGA) dataset showed that higher *SOCS3*, *VEGFA*, and *TEK* expression levels are observed in GBM cases with wildtype (WT) *IDHs* (Figure 4A–C). To further test the relationship between *IDH* mutations and *SOCS3*, *VEGFA*, and *TEK* methylation, we grouped primary glioma and GBM cases in the TCGA-LGG/GBM dataset into cohorts with mutated (MT) and WT *IDH1*. These cases were aligned with decreasing β values of *SOCS3* methylation, representing the percentage of DNA methylation ratio in each cohort. The methylation status of *VEGFA* and *TEK* of those aligned cases were then presented in the heatmap analysis. We found that *IDH1* mutation is associated with hypermethylation of *SOCS3, VEGFA*, and *TEK*, indicating that the expression of these genes is downregulated epigenetically in *IDH1* MT GBM cases (Figure 4E). Additionally, we found that the methylation status of *SOCS3* and *VEGFA* is closely correlated in both *IDH1* WT and MT groups (Figure 4E). This is consistent with the Pearson correlation analysis, which showed that the expression of *SOCS3* and *VEGFA* has a strong positive association (R-value = 0.69, *p* < 2.2 × 10^−16^) in glioma (Figure 4D). These results suggested that the favorable prognosis of *IDH1*-mutated patients may partially contribute to *SOCS3*, *TEK*, and *VEGFA* hypermethylation and the subsequent downregulation of angiogenic activities. 

### 2.5. Radiogenomics of GBM Samples with Differential SOCS3 Expression Levels

Perfusion-weighted magnetic resonance imaging (PW-MRI), including dynamic contrast-enhanced MRI (DCE-MRI) and dynamic susceptibility contrast-MRI (DSC-MRI), can effectively measure features of GBM vasculature, such as blood volume and blood flow, as well as microvascular leakage [24]. To evaluate the relevance of *SOCS3* expression with radiographical imaging features of GBM, we studied all of the TCGA-GBM cases that had both *SOCS3* expression and MRI perfusion data available. Eight GBM cases were identified. We found that samples with *SOCS3* expression levels higher than the median cutoff value (TCGA-06-0645; TCGA-06-0646; TCGA-06-0878) had statistically significant higher perfusion intensity relative to samples with *SOCS3* expression levels lower than the median cutoff value (TCGA-06-0178; TCGA-14-1829; TCGA-06-0174) based on 3D quantification of DSC-MRI blood flow (BF) signal intensity (*p* < 0.05) (Figure 5A,C). Quantification of the DCE-MRI images showed average area under contrast curve (AUC) values of 19,488 and 3440 for *SOCS3* higher than the median (TCGA-06-5412) and lower than the median (TCGA-06-2570) expression cases, respectively (Figure 5B). These data suggested that elevated *SOCS3* expression is positively associated with enhanced perfusion in GBM, highlighting the potential of integrating *SOCS3* expression levels into radiogenomic studies to achieve a more accurate noninvasive molecular profiling strategy for GBM based on perfusion MRI workup. 

### 2.6. Indication of Angiogenesis Inhibitors for GBM with Differential SOCS3 Expression

Anti-angiogenic therapy with bevacizumab (BVZ) is the last-line treatment for GBM patients following the failure of radiotherapy, temozolomide, and lomustine [25]. Although BVZ is the most common treatment option following recurrence of GBM, there are substantially different tumor responses with a lack of effective biomarkers for patient selection. To this end, we studied whether *SOCS3* expression could be used as a biomarker for the selection of patients who are more likely to have a better response toward BVZ treatment. In the Ivy-GAP dataset, we identified eight patients treated with BVZ who also had RNA-Seq data of *SOCS3* expression at anatomical locations of a cellular tumor. Patients who had average *SOCS3* expression Z-scores > 0 were considered as a *SOCS3* High expression group. Patients with *SOCS3* expression Z-scores < 0 were identified as a *SOCS3* Low expression group. T1 post-Gad MRI images were studied before and after BVZ treatment (Figure 6). Each patient received at least two doses of BVZ, taken two to three times per month (Appendix A). We found that patients with higher expression levels of *SOCS3* (Z-Score > 0) generally had a better response toward angiogenesis inhibition, as evidenced by decreased or stabilized enhancement after BVZ treatment (Figure 6B, arrows). However, all patients in the *SOCS3* Low expression group still experienced GBM progression within the BVZ treatment period time (Figure 6A). Therefore, elevated expression of *SOCS3* might be used as a marker for selecting patients who are more likely to benefit from BVZ treatment.

## 3. Discussion

In 2016, the World Health Organization (WHO) reclassified central nervous system (CNS) tumors by integrating molecular/genetic criteria into histological diagnostics, which produced a paradigm shift in precise diagnosis, personalized therapeutics, and prognostic factor-guided treatment decisions [15]. To this end, our results provide a simple three-gene transcriptomic signature of *SOCS3* plus the angiogenic genes *VEGFA* and *TEK* as an independent prognosis factor for all of the GBM subclasses. SOCS3 is a member of eight molecules (SOCS1–7 and CIS) that dampen the signal transducer and activator of transcription protein (STAT) activation by direct interactions with Janus kinases (JAKs) and STAT (JAK-STAT)-activating receptors [26]. SOCS3 is also a substrate recruiting protein in the cullin5-RING E3 ligase (CRL5) complex that poly ubiquitinates substrates for the ubiquitin-proteasomal system (UPS)-mediated degradation [27]. SOCS3 mediates ubiquitination of a plethora of substrates critically involved in inflammatory response, including NF-κB family member p65/RelA, VHL, the granulocyte colony-stimulating factor (G-CSF) receptor, and the insulin receptor substrate-1 (IRS-1) [28,29,30]. Collectively, elevated *SOCS3* expression at perivascular and pseudopalisading cells around necrosis (PAN) likely indicates the local inflammatory response that triggers its transcription. 

Our data showed an inverse correlation between protein levels of cullin5 and VHL, suggesting a potential role of SOCS3 in the neovascularization of GBM since VHL is a critical anti-angiogenic protein [31]. This conclusion is further substantiated by increased vascularization in GBM samples with enhanced SOCS3 protein immunoreactivities and elevated *SOCS3* expression levels. Increased neovascularization is often observed in GBM cases with a poor prognosis, which often possess exuberant neovascularization that includes disorganized, irregular, and tortuous vessels [15]. The abnormal vascular structure is reflected by increased microvascular permeability and perfusion that can be demonstrated in perfusion-weighted magnetic resonance imaging (PW-MRI) techniques, such as dynamic contrast-enhanced MRI (DCE-MRI) and dynamic susceptibility contrast-MRI (DSC-MRI) [24]. Based on the TCGA dataset, we found that patients with elevated *SOCS3* expression often had enhanced perfusion intensities compared with patients bearing lower expression levels of *SOCS3*. Altogether, these data could potentially integrate radiographic features into molecular/genomic expression patterns of GBM, linking specific imaging features with noninvasive genomic profiling. 

The therapeutic response of GBM toward bevacizumab (BVZ) varied substantially among patients. Therefore, an effective biomarker for patient selection is urgently needed. We found that patients with elevated *SOCS3* expression are more likely to respond to BVZ. Patients with high *SOCS3* expression may represent a group of GBM cases that are more dependent on neovascularization for recurrence and progression. However, the response to BVZ in this study was accessed by radiological features characterized by reduced or stabilized enhancement during the treatment timeframe. All eight patients selected for the study eventually succumbed to the disease shortly after the BVZ regimen. This is consistent with clinical trials that showed BVZ did not improve overall survival (OS) in GBM [32]. As the last line of treatment for recurrent GBM patients who have failed radio- and chemotherapies, initiation of BVZ might be too late to make a difference regarding OS. Accordingly, it is promising that molecular profiling of *SOCS3* expression levels in GBM samples promptly after surgical resection might be used to select a group of GBM patients who might benefit from earlier initiation of BVZ therapy. However, as a retrospective study, this study included a limited number of patients who received BEV treatment and at same time had available *SOCS3* RNA-Seq results. Furthermore, most of the patients treated with BEV for a limited period of time (Appendix A). Therefore, the results presented suggesting *SOCS3* as a biomarker for the selection of patients who may benefit from BEV treatment should be interpreted with caution.

The underlying molecular mechanism of *IDHs* mutations on angiogenesis remains complicated and controversial, as both HIF1α stabilization and decreased *HIF1A* transcription have been reported with *IDH1* mutations [22,33,34]. We found that *SOCS3* and *VEGFA* expressions are downregulated in *IDH1* mutation cases, correlating with hypermethylation of *SOCS3* and *VEGFA* and the subsequent decreased transcriptional activities. However, *IDH* mutations are still relatively rare among GBM cases. In conclusion, the three-gene transcriptome signature of *SOCS3*, *VEGFA*, and *TEK* may play a significant role in prognosis, treatment planning, and radiogenomics profiling in GBM, regardless of *IDH* mutation status. 

## 4. Materials and Methods 

### 4.1. Public Datasets

The primary databases of samples were derived from TCGA-GBM, which includes 540 GBM patients with information for copy number, RNA-Seq, and DNA methylation status. The Chinese Glioma Genome Atlas Network (CGGA) provides information on grading, RNA-Seq, and *IDHs* mutations of 1962 glioma patients (http://www.cgga.org.cn/) [35]. The primary clinicopathological samples with an anatomic transcriptional atlas were collected from the Ivy Glioblastoma Atlas Project (Ivy-GAP), and The Cancer Imaging Archive (TCIA) provides radiological phenotypes of 262 GBM patients [19,36]. The TCGA low-grade glioma (LGG) and glioblastoma (GBM) (LGG/GBM) provides information on DNA methylation, RNA-Seq, and *IDHs* mutations of 1153 glioma patients. Expression levels of genes in the normal tissue were based on the Genotype-Tissue Expression (GTEx) project (https://www.gtexportal.org/home/). All these samples were collected with informed consent.

### 4.2. Anatomical Mapping of SOCS3 and Angiogenic Genes’ Expression

Ivy-GAP provides a 41-patient cohort whose tumor samples were evaluated based on anatomic features classified as leading edge (LE), infiltrating tumor (IT), cellular tumor (CT), pseudopalisading cells around necrosis (PAN), and microvascular proliferation (MVP). All the samples included in this study were classified with hematoxylin and eosin (H&E) staining histology features as a reference. Major markers involved in angiogenesis—*ANGPT1*, *ANGPT2*, *FLT1*, *PECAM1*, *TEK*, *TIE1*, *VEGFA*, *NRP1*, *KDR*, and *SOCS3*—were included. In situ hybridization (ISH) for *VEGFA* (*n* = 9) and *SOCS3* (*n* = 21) and the corresponding H&E staining data were also derived from the Ivy-GAP. ISH analysis for *TEK* was derived from a normal mouse brain generated by the Allen Institute. 

### 4.3. Radiographic Response to Bevacizumab (BVZ)

Clinical cases in the Ivy-GAP dataset that were treated with BVZ were selected based on the Z-score of *SOCS3* expression levels. Patients with an average Z-score > 0 were grouped as a *SOCS3* High expression cohort (subject ID: W11, W4, W3, and W43) and those with a Z-score < 0 were considered as a *SOCS3* Low expression cohort (subject ID: W26, W31, W8, and W48). Each patient received at least two doses of BVZ in total, taken two to three times per month. Detailed information on the dosages of BVZ as well as prior therapeutics and demographics of these patients is available at Appendix A and can be also found at http://glioblastoma.alleninstitute.org/. Corresponding post-gadolinium (post-Gad) T1-weighted axial images before BVZ and after BVZ treatment are presented.

### 4.4. Immunohistochemistry and Proteomic Analysis

The Human Protein Atlas provides immunohistochemistry (IHC) analyses of *SOCS3* in GBM samples (https://www.proteinatlas.org/). The anti-SOCS3 antibody CAB012220 was used for the IHC study. Four GBM patients were included in the IHC study. Proteomic study data used in this publication were generated by the National Cancer Institute Clinical Proteomic Tumor Analysis Consortium (CPTAC) (https://cptac-data-portal.georgetown.edu/). Samples that had data of VHL expression were selected (*n* = 99). Mass spectrometry analysis was conducted using the 11-plexed isobaric tandem mass tags (TMT-11). Relative protein abundance is presented as Log2(ratio). GBM cases were aligned with decreasing cullin5 protein levels. Corresponding protein levels of VHL of aligned samples are color-coded in the heatmap analysis. 

### 4.5. Analysis of Expression Levels and Methylation Status of SOCS3, VEGFA, and TEK

The DNA methylation of *SOCS3*, *VEGFA*, and *TEKI* in glioma was analyzed on the UCSC-Xena platform (https://xena.ucsc.edu/). The methylation was examined using Infinium HumanMethylation450 BeadChip with the level of each CpG site represented by the beta (β) value. Patients in the TCGA-LGG/GBM dataset were first grouped into *IDH1* wildtype (WT; *n* = 417) and *IDH1* mutated (MT; *n* = 412) cohorts. Samples in these cohorts were then arranged with decreasing *SOCS3* methylation β-values. The corresponding methylation statuses of *VEGFA* and *TEK* of the aligned samples were then color-coded based on their β values. The heatmap analysis for the expression of *SOCS3*, *VEGFA*, and *TEK* in GBM was also carried out in UCSC-Xena. Briefly, patients from the TCGA-GBM dataset were first aligned based on decreasing *SOCS3* expression levels (*n* = 173). Corresponding expression levels of *VEGFA* and *TEK* of these aligned cases were color-coded. All the RNA-sequencing was carried out using Illumina HiSeq. Gene expression was quantified by log2(norm_count+1), where norm_count refers to RNA-Seq expression estimation by expectation–maximization (RSEM) normalized count.

### 4.6. Multi-Gene Prognostic Index Analysis

Survival analysis based on the impact of the multi-gene prognostic index (PI) was conducted by GBM-BioDP (https://gbm-biodp.nci.nih.gov/) under Verhaak Core [20]. Briefly, GBM was classified into proneural (P), neural (N), classical (C), and mesenchymal (M) subtypes based on gene expression patterns [20]. The Cox proportional hazards model was constructed with gene expressions of *SOCS3*, *TEK*, and *VEGFA* as covariates. The Logrank test was used for calculations of *p*-values. 

### 4.7. Perfusion-Weighted MRI

Clinical cases in the TCGA-GBM dataset with higher (TCGA-06-0645; TCGA-06-0646; TCGA-06-087; TCGA-06-5412) and lower than the median (TCGA-06-0178; TCGA-14-1829; TCGA-06-0174; TCGA-06-2570) expression levels of *SOCS3* were identified. Briefly, cases with available data on *SOCS3* expression levels were identified and sub-grouped into *SOCS3* High and *SOCS3* Low cohorts based on the *SOCS3* median expression level. Then cases that have quantifiable imaging data for perfusion-weighted magnetic resonance imaging (PW-MRI) analysis, including dynamic contrast-enhanced MRI (DCE-MRI) and dynamic susceptibility contrast-MRI (DSC-MRI) analysis in the TCIA dataset were selected (https://www.cancerimagingarchive.net/). 3D blood flow (BF) signal intensities within regions-of-interests (ROIs) were quantified automatically with NordicICE 4.2.0. Corresponding pre-gadolinium (pre-Gad) T1-weighted axial images were derived from TCIA. 

### 4.8. Statistical Analysis 

Statistical analysis was performed using GraphPad Prism 6.0 software. Pearson correlation analysis between *VEGFA* and *SOCS3* expression levels in primary glioma was carried out automatically in the CGGA dataset. Kaplan–Meier plot analysis for the 3-gene (*SOCS3*, *VEGFA*, and *TEK*)-based prognosis is based on prognostic index (PI), which is the linear component of the Cox model: PI = β_1_x_1_ + β_2_x_2_ + … + β_p_x_p_. Samples were then stratified by the PI and grouped by median value with higher values for higher risk. Equal number of samples were assigned to each group when using median PI value as cutoff. The Logrank test was used for calculations of *p*-values for the resulting 2 cohorts. All statistical tests were 2-sided, and *p*-values smaller than 0.05 were considered statistically significant.

## 5. Conclusions

This study identified a simple functional transcriptome signature of *SOCS3* plus angiogenesis markers *VEGFA* and *TEK* for GBM prognosis. Elevated *SOCS3* expression levels correlate with increased *VEGFA* expression levels and GBM neovascularization. Postsurgical quantification of *SOCS3* may facilitate the identification of GBM patients who will have better therapeutic response towards angiogenesis inhibitors.

## Figures and Tables

**Figure 1 cancers-13-01013-f001:**
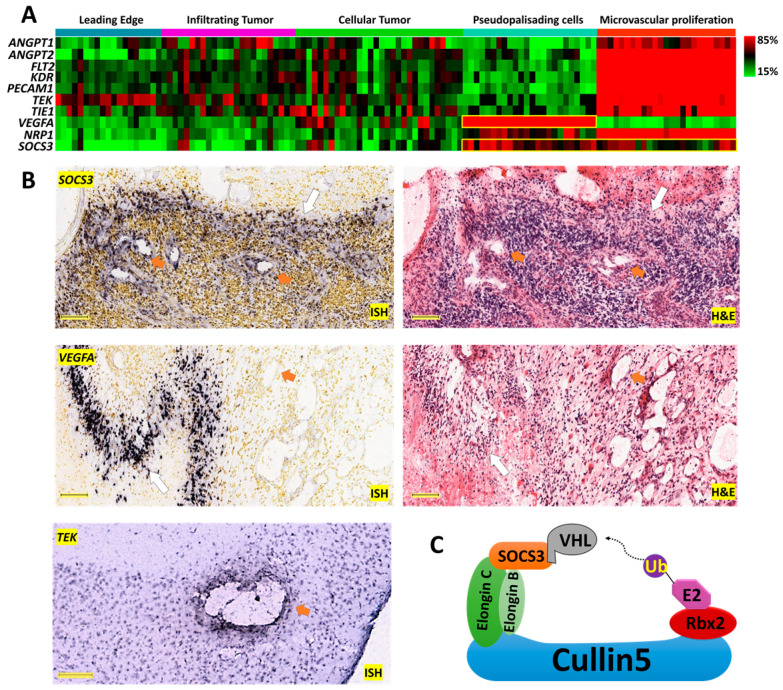
Anatomical mapping of *SOCS3* and angiogenic genes in glioblastoma (GBM) clinical samples. (**A**) Expression levels of angiogenesis markers *ANGPT1*, *ANGPT2*, *FLT1*, *PECAM1*, *TEK*, *TIE1*, *VEGFA*, *NRP1*, and *KDR* are mapped on corresponding anatomical structures. The yellow squares highlight the anatomical expression patterns of *VEGFA* and *SOCS3*. (**B**) In situ hybridization (ISH) and hematoxylin and eosin (H&E) analysis of *SOCS3* and *VEGFA* in GBM patients, Scale bar: 200 μM. ISH analysis of TEK in normal murine cranial tissue. Scale bar: 150 μM. Image credit: Ivy-GAP, Allen Institute. Orange arrows indicate blood vessels. White arrows indicate pseudopalisading cells around necrosis. (**C**) Schematic overview of the CRL5 E3 ligase. SOCS3 in the CRL5 can recruit von Hippel–Lindau (VHL) for polyubiquitination and degradation.

**Figure 2 cancers-13-01013-f002:**
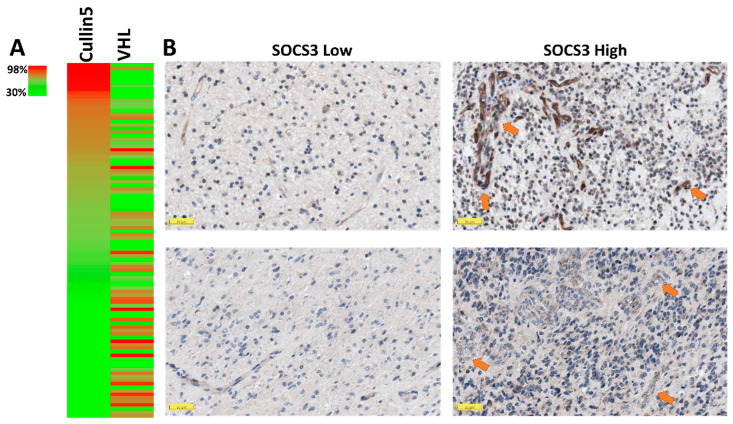
Clinicopathological study of SOCS3 protein in GBM. (**A**) Heatmap analysis of proteins cullin5 and VHL in GBM. (**B**) Immunohistochemistry staining for SOCS3 in 4 GBM cases. Arrows indicate blood vessels. Image credit: Human Protein Atlas. Scale bar: 50 μM.

**Figure 3 cancers-13-01013-f003:**
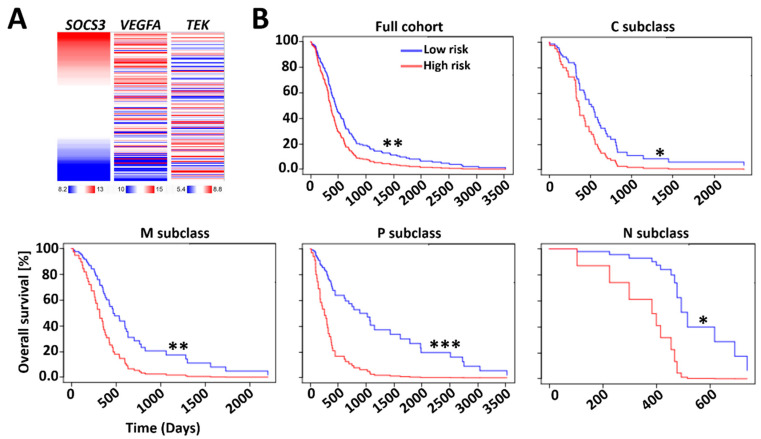
Survival analysis based on the impact of the multi-gene prognostic index (PI). (**A**) Heatmap analysis of expression levels of *SOCS3*, *VEGFA*, and *TEK* was conducted on the UCSC Xena platform based on The Cancer Genome Atlas (TCGA)-GBM dataset (*n* = 197). (**B**) Survival analysis based on hazard ratio (HR) was conducted via the GBM-BioDP (https://gbm-biodp.nci.nih.gov/). GBM was classified into proneural (P), neural (N), classical (C), and mesenchymal (M) subtypes based on gene expression patterns [20]. The stratification of the three-gene signatures for full cohort, P, M, and C subclasses is based on increasing PI levels of 1Half vs. 2Half; N subclass is based on 1Qt vs. 4Qt for stratification. *: *p* < 0.05; **: *p* < 0.01; ***: *p* < 0.001.

**Figure 4 cancers-13-01013-f004:**
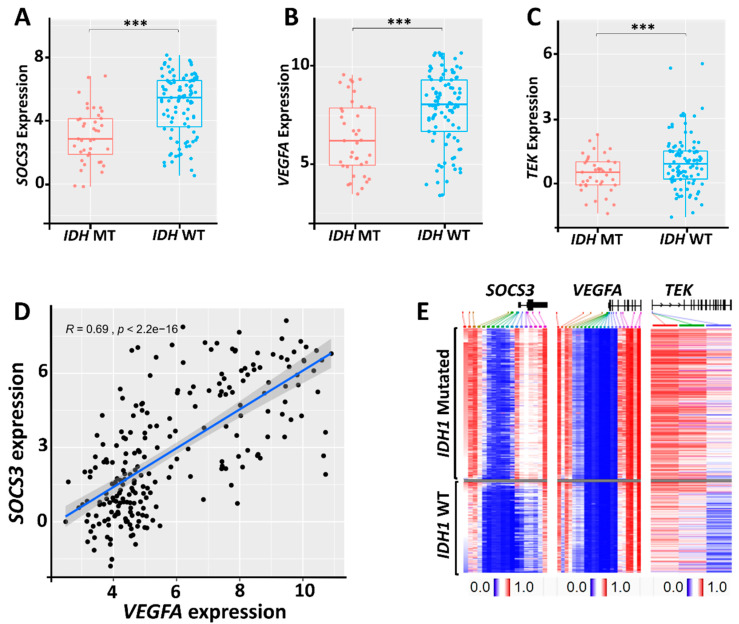
Correlation of *SOCS3* and angiogenic genes’ expression levels with *IDH* mutation status. (**A**–**C**) Expression levels of corresponding genes in *IDHs* mutated (MT) and *IDHs* wildtype (WT) GBM. (**D**) Pearson correlation analysis of *VEGFA* and *SOCS3* expression in primary glioma. (**E**) Heatmap analysis of *SOCS3, TEK*, and *VEGFA* genes’ methylation with *IDH1* mutation status based on TCGA-LGG/GBM dataset. Gray area indicates cases with methylation data that are not available. ***: *p* < 0.001.

**Figure 5 cancers-13-01013-f005:**
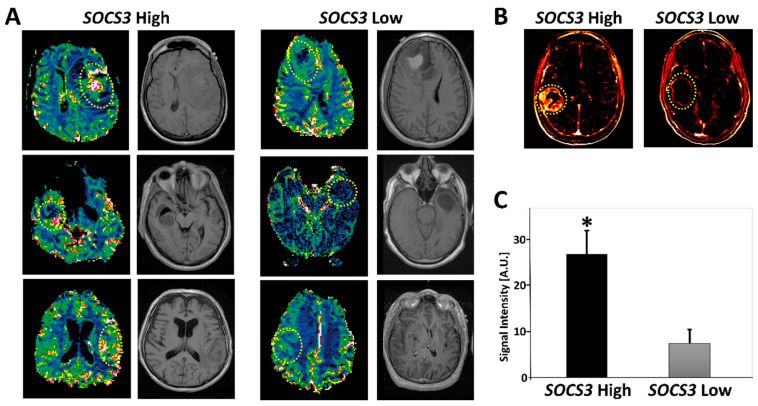
Radiogenomics of *SOCS3* expression in GBM. (**A**) Representative dynamic susceptibility contrast-MRI (DSC-MRI) images with higher than the median *SOCS3* expression and lower than the median *SOCS3* levels. Circles represent regions of interest (ROIs). (**B**) Dynamic contrast-enhanced (DCE)-MRIs of higher than the median (TCGA-06-5412) and lower than the median (TCGA-06-2570) *SOCS3* expressions. (**C**) Quantification of 3D tumor volume perfusion intensity in groups with differential *SOCS3* expression levels. *: *p* < 0.05.

**Figure 6 cancers-13-01013-f006:**
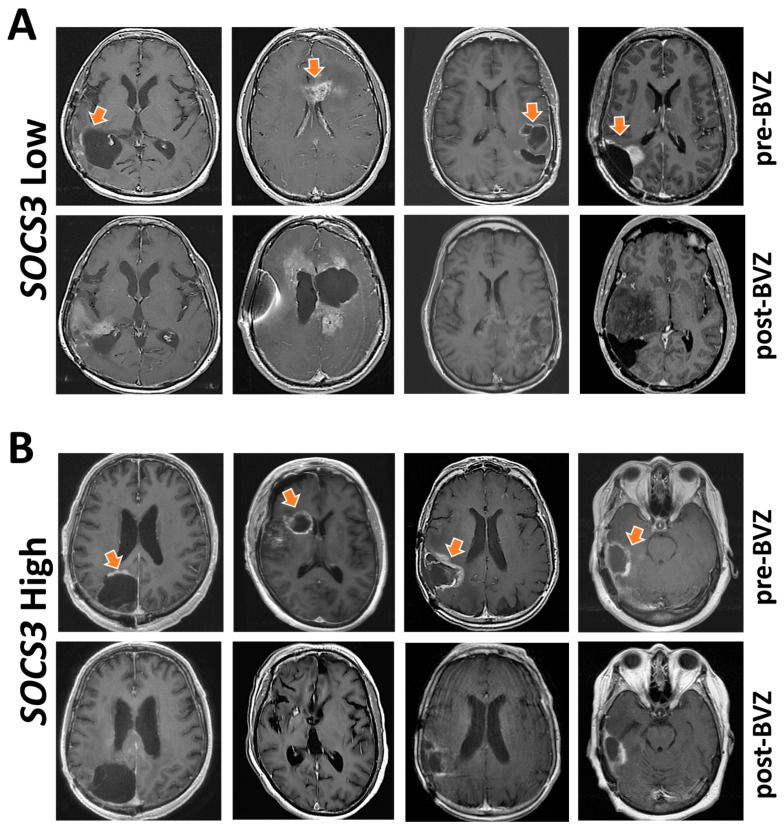
*SOCS3* expression levels and angiogenesis inhibition. Eight patients were selected from the Ivy Glioblastoma Atlas Project (Ivy-GAP) dataset. (**A**) Four patients who had average *SOCS3* expression Z-scores > 0 were considered as a *SOCS3* High expression group. (**B**) Another group of four patients with *SOCS3* expression Z-scores < 0 were identified as a *SOCS3* Low expression group. T1 post-Gad MRI images were studied before and after BVZ treatment. Each patient received at least two doses of BVZ. Orange arrows indicate radiological features related to GBM progression that decreased post-BVZ treatment in the *SOCS3* High expression group.

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
