# Peer review of "Identification of Novel Transcriptome Signature as a Potential Prognostic Biomarker for Anti-Angiogenic Therapy in Glioblastoma Multiforme"

_cancers, 2021, doi:10.3390/cancers13051013_

Round 1
Reviewer 1 Report
The research Article ISSN 2072-6694 entitled " Transcriptome signature of SOCS3 and angiogenic markers for glioblastoma multiforme prognosis and molecular profiling" by Shuhua Zheng and Wensi Tao is a review paper aiming to identify a simple functional transcriptome signature of SOCS3 plus angiogenesis markers VEGFA and TEK for GBM prognosis. Elevated SOCS3 expression may guide the selection of patients who have increased GBM neovascularization and sensitization towards angiogenesis inhibition.
The paper topic is interesting, the data has been well presented, but in my opinion, the results discussed in the document do not seem relevant enough for a complete, conclusive and useful review in a prestigious journal such Cancers.
Author Response
Dear reviewer, we appreciate your positive comments in our manuscript. Please note all the changes that we had made in the revised manuscript. The manuscript presented in a Research article focusing on identification of biomarkers for GBM prognosis and identification of GBM patients who may benefit from angiogenesis inhibitors. We do hope that all the revisions will make the article in better position for possible publication in the journal.
Thank you!
Reviewer 2 Report
Zheng and Tao took into account an impressive number of glioma and GBM cases. They identified a three-gene transcriptome signature (SOCS3, VEGFA and TEK) that connect GBM overall prognosis with genes’ expression, correlating radiographical features of perfusion imaging with SOCS3 levels. Their work pointed out SOCS3 as a marker for patients who may benefit from early initiation of angiogenesis inhibitors.
In my opinion, the paper is well written and presented; also, the data are interesting. It could be relevant to make an additional evaluation, dividing all gliomas considered based on their classification and then look at the SOCS3 levels, to see if there're differences among them and whether some forms might have a clearer correlation.
Minor points:
Fig 2A: I noticed a different background of one image of SOCS3 High. Why?
Fig 2A: Please make the scale bar more visible in the picture.
Fig 6: Please note that part of the figure (on the right) was cut. Please use orange arrows also to point out the tumor in the above part of the figure.
Reviewer 3 Report
In this manuscript Shuhua Zheng and Wensi Tao, by analysing glioma and glioblastoma multiforme (GBM) cases across four independent datasets, detected SOCS3, VEGFA, and TEK gene transcriptome signature associated with GBM prognosis. Moreover, the authors correlated radiographical features of perfusion imaging with the expression level of SOCS3, and showed a better therapeutic response to bevacizumab (BVZ) in GBM patients with high SOCS3 expression.
In the complex the manuscript contains some important weaknesses that need to be addressed before a possible publication.
1) The authors should ameliorate dramatically the quality of Figure 1 B and should specify the scale bar. Moreover, the authors should add an in situ hybridization (ISH) analysis of TEK. The authors should further indicate how many GBM cases are represented by in situ hybridization analysis showed in Figure 1 B.
2) The authors should improve the quality of Figure 1C.
3) In the Figure 2B the authors show immunohistochemical staining for SOCS3 of GBM tissues from 4 cases. Among these tissues, two should represent GMB tissues with low SOC3 immunostaining whereas the other two samples should represent GMB tissues with high SOCS3 immunostaining. First of all, two samples are too few to infer a correlation between SOCS3 immunostaining and neovascularization. Moreover, the immunostaining of cancer tissues with low levels of SOCS3 looks like negative controls, whereas one SOCS3 immunostained tissue section, from GBM sample with high level of SOCS3, resembles a GBM tissue characterized by low level of SOCS3.
4) Replace the figures 6A and 6B because they present a cutted text.
5) Line 38, “with IDH-mutated with 1p/19q codeletion often predict favorable prognosis”: the authors should clarify this sentence.
6) Line 224-225, “……and elevated SOCS3 expression levels in the H&E staining analysis”: the authors should clarify this sentence.
Reviewer 4 Report
The paper describes scientific work with the aim to identify biomarker(s) for an early benefit from Bevacizumab (BEV) in different subtypes of Glioblastoma. Indeed, a very important clinical question.
Comments:
- Title: is it on purpose, that the title does not reveal the main hypothesis of the study ?
- Introduction: could the hypothesis of the study be formulated more concisely ? Please comment on why conclusions of the study do appear in the introduction part (line 66-70) ?
- Material & Methods: despite the fact, that the most important Glioma datasets were consulted (CGGA, TCGA, Ivy-GAP, TCIA, CPTAC-GBM) for this study, we do not learn how samples were selected. There is only little evidence provided on radiografic and clinical outcomes of GBM patients during a short BEV therapy (N=8 from the Ivy-GAP dataset). Detailed information on the dosage of BEV as well as prior therapeutics and demographics of the patients is said to be available at http://glioblastoma.alleninstitute.org/. However detailed clinical data such a concomitant steroid use etc. cannot be retrieved, since the domain is password protected. For the very few 8 patients, clarifying informations should be transferred into the manuscript. The Statistic part should be expanded.
- Results: the clinical part of the project is based on a very small patient number. The follow-up period for the few clinical cases is extremely short (about 1 months, at least 2 doses of BEV). Therefore conclusions cannot be made and any correlation of elevated SOCS3 expression levels with improved tumor response towards BEV therapy is hypothesis generating at best.
Figure 3: Please provide patient numbers for „full cohort“ (N=173 ?). Please define „low and high risk“
Figure 6: needs time scales. Angles of images should be congruent. Was there a surgical intervention during BEV in the second patient of group A.
Three genes were identified SOCS2, VEGFA and TEK: how independent are these prognostic and predictive biomarkers ?
- Conclusions: the authors conclusion „Our study identified SOCS3 expression levels as a potential selection marker for patients who may benefit from early initiation of angiogenesis inhibitors“ should be based on more robust clinical evidence.
Round 2
Reviewer 3 Report
The manuscript has been substantially improved
Author Response
Dear reviewer, we had checked the spells of the revised manuscript again. Thank you for giving us another opportunity to improve our manuscript.
Reviewer 4 Report
Many thanks for a careful revision of the manuscript
Based on the data presented, only preliminary conclusion is possible. The title should therefore be formulated more cautiously.
e.g. Identification of a Potential Novel Transcriptome Signature as Prognostic Biomarker for Anti-angiogenic Therapy in Glioblastoma Multiforme
page 9, line 216:
Conclusions here have to be attenuated:
"Therefore, elevated expression of SOCS3 can might be used as a marker for selecting patients who are more likely to benefit from BVZ treatment".
page 12, line 269:
"Accordingly, it is promising that molecular profiling of SOCS3 expression levels in GBM samples promptly after surgical resection can might be used to select the a group of GBM patients who will might benefit from the OS perspective" from earlier initiation of BVZ therapy.
Author Response
Q1: Many thanks for a careful revision of the manuscript
Based on the data presented, only preliminary conclusion is possible. The title should therefore be formulated more cautiously.
e.g. Identification of a Potential Novel Transcriptome Signature as Prognostic Biomarker for Anti-angiogenic Therapy in Glioblastoma Multiforme
R1: Dear reviewer, thank you again for giving us another opportunity to improve our manuscript. The title of manuscript is changed to: 'Identification of a Novel Transcriptome Signature as a Potential Prognostic Biomarker for Anti-angiogenic Therapy in Glioblastoma Multiforme'.
Q2: page 9, line 216:
Conclusions here have to be attenuated: "Therefore, elevated expression of SOCS3 can might be used as a marker for selecting patients who are more likely to benefit from BVZ treatment".
R2: In the revised manuscript, Line 194: ‘Therefore, elevated expression of SOCS3 can be used as a marker for selecting patients who are more likely to benefit from BVZ treatment.’ is changed to: ‘Therefore, elevated expression of SOCS3 might be used as a marker for selecting patients who are more likely to benefit from BVZ treatment.’
Q3: page 12, line 269:
"Accordingly, it is promising that molecular profiling of SOCS3 expression levels in GBM samples promptly after surgical resection can might be used to select the a group of GBM patients who will might benefit from the OS perspective" from earlier initiation of BVZ therapy.
R3: In the revised manuscript, Line 242: ‘Accordingly, it is promising that molecular profiling of SOCS3 expression levels in GBM samples promptly after surgical resection can be used to select the group of GBM patients who will benefit from the OS perspective resulting from earlier initiation of BVZ’ is changed to ‘Accordingly, it is promising that molecular profiling of SOCS3 expression levels in GBM samples promptly after surgical resection might be used to select the group of GBM patients who might benefit from earlier initiation of BVZ therapy’.